# Preliminary Identification of Candidate Genes Related to Survival of Gynogenetic Rainbow Trout (*Oncorhynchus mykiss*) Based on Comparative Transcriptome Analysis

**DOI:** 10.3390/ani10081326

**Published:** 2020-07-31

**Authors:** Konrad Ocalewicz, Artur Gurgul, Marcin Polonis, Stefan Dobosz

**Affiliations:** 1Department of Marine Biology and Ecology, Institute of Oceanography, Faculty of Oceanography and Geography, University of Gdansk, M. Piłsudskiego 46 Av, 81-378 Gdynia, Poland; marcin.polonis@o2.pl; 2Centre for Experimental and Innovative Medicine, University of Agriculture in Kraków, Rędzina 1c, 30-248 Kraków, Poland; Artur.Gurgul@urk.edu.pl; 3Department of Salmonid Research, Inland Fisheries Institute in Olsztyn, Rutki, 83-330 Żukowo, Poland; s.dobosz@infish.com.pl

**Keywords:** egg quality, gynogenesis, maternal RNA, rainbow trout, transcriptome

## Abstract

**Simple Summary:**

Gynogenetic doubled haploid zygotes in fish are produced by the activation of eggs with irradiated spermatozoa, followed by the exposure of the activated eggs to the temperature or high hydrostatic pressure (HHP) shock that prevents the first cell cleavage. Variation in the survival rates of the gynogenetic embryos developing in eggs originating from different females have been reported in several studies. As eggs of different origin show altered ability for the gynogenetic development, it was tempting to assume that the ova showing such a vast difference in potential for gynogenesis may have also had different maternal gene expression profiles. To verify these hypotheses, we compared the next-generation sequencing data of the maternal mRNA collected from rainbow trout eggs showing varied ability to develop after activation with UV-irradiated spermatozoa. Eggs originating from females whose gynogenetic offspring had the highest survival showed an increased expression of 46 genes associated among others with cell survival, migration and differentiation, triglyceride metabolism, the biosynthesis of polyunsaturated fat, and early embryogenic development, 5S RNA binding, embryonic neurogenesis and tissue modelling during development as well as senescence and aging, among others. Such genes may be considered as potential candidate genes whose expression enable efficient gynogenesis in rainbow trout.

**Abstract:**

In the present research, the eggs from four rainbow trout females were used to provide four groups of gynogenetic doubled haploids (DHs). The quality of the eggs from different clutches was comparable, however, interclutch differences were observed in the gynogenetic variants of the experiment and the survival of DH specimens from different groups varied from 3% to 57% during embryogenesis. Transcriptome analysis of the eggs from different females exhibited inter-individual differences in the maternal genes’ expression. Eggs originating from females whose gynogenetic offspring had the highest survival showed an increased expression of 46 genes when compared to the eggs from three other females. Eggs with the highest survival of gynogenetic embryos showed an up-regulation of genes that are associated with cell survival, migration and differentiation (tyrosine-protein kinase receptor TYRO3-like gene), triglyceride metabolism (carnitine O-palmitoyltransferase 1 gene), biosynthesis of polyunsaturated fat (3-oxoacyl-acyl-carrier-protein reductase gene), early embryogenic development (protein argonaute-3 gene, leucine-rich repeat-containing protein 3-like gene), 5S RNA binding (ribosome biogenesis regulatory protein homolog) as well as senescence and aging (telomerase reverse transcriptase, TERT gene), among others. Positive correlation between the genotypic efficiency and egg transcriptome profiles indicated that at least some of the differentially expressed genes should be considered as potential candidate genes for the efficiency of gynogenesis in rainbow trout.

## 1. Introduction

Fish egg quality is described as the ability of egg to be fertilized and subsequently to develop into a normal embryo and larvae [1]. Egg quality is influenced by many internal and external factors, such as maternal age and genetics, photoperiod regimes, hormonal stimulation, stress, but also husbandry management including handling and feeding [2]. Moreover, preliminary or delayed collection and/or the extended time of eggs storage before insemination may result in post-ovulatory aging and the consequent decrease in the egg developmental competence [3,4,5].

The quality of eggs is also one of the most important factors in the successful application of chromosome set manipulations in fish including gynogenesis, androgenesis or triploidization [6] as such manipulations require the exposure of eggs to harmful treatments including irradiation (UV and ionizing) and sublethal physical shocks. Haploid gynogenetic and androgenetic zygotes are produced by the activation of eggs with irradiated spermatozoa and the insemination of irradiated eggs with non-treated sperm, respectively. The duplication of maternal or paternal chromosomes and the production of gynogenetic and androgenetic doubled haploids (DHs), respectively, may be achieved by the exposure of the haploid embryos to the temperature or high hydrostatic pressure (HHP) shock around the prophase of the first mitosis that prevents the first cell cleavage [7]. In turn, the application of physical shock to fish eggs shortly after activation with normal or irradiated (UV) spermatozoa disturbs the second meiotic division and prevents the release of the second polar body, which results in the development of the triploid and diploid meiogynogenetic embryos, respectively. As several experiments showed that the listed above the manipulations may decrease developmental competence of the treated eggs [8], it is highly recommended to use only the best quality gametes for gynogenesis, androgenesis and triploidization.

The early stages of vertebrate embryonic development completely rely on the maternal mRNA deposited in the eggs during oogenesis. In fish, maternal mRNA translated upon egg activation governs a series of synchronous cell divisions before the initiation of the zygotic genome activation (ZGA), which usually takes place during mid-blastula transition (MBT) that occurs in rainbow trout 2 days post-fertilization [9]. Several experiments performed on the model and aquaculture fish species exhibited that maternal gene expression in eggs may be altered by internal and external factors acting during oogenesis and the quality of eggs is paralleled with their transcriptome profiles [10]. Transcriptomic studies using the next-generation sequencing approach showed that the differences in maternal mRNA abundance in zebrafish eggs were bigger between clutches from different mothers than within the clutches [11]. In turn, in rainbow trout, greater differences in the maternal gene expression were observed between the eggs from different clutches than between the HHP-treated and untreated eggs from the same clutch [12]. Both reports indicate that differences in the expression of the maternal genes primarily result from differences between mothers. In the androgenetic and gynogenetic experiments, the differences in survival rates of embryos developing in eggs originating from different females have been reported by several authors [13,14,15], among others. Similar events were also observed in the gynogenetic experiment described here, where significant differences in the survival rates between gynogenetic DH rainbow trout developing in eggs originated from four gamete donors were shown. The eggs of one of the females exhibited great potential for development after activation with UV-treated spermatozoa, manifested by the unusually high survival (>50%) of gynogenetic specimens. As the eggs from other females show several times lower ability for gynogenetic development, it was tempting to assume that the ova showing such a vast differences in potential for gynogenesis may have also had different maternal gene expression profiles. If so, then the comparative trancriptomic analysis of eggs from different clutches exhibiting varied competence for gynogenesis might help to identify the potential genes related to gynogenetic development in rainbow trout. To verify this hypothesis and to find the candidate genes involved in the gynogenetic development in rainbow trout we compared the next generation sequencing data of the maternal mRNA collected from eggs showing varied ability to develop after activation with UV-irradiated spermatozoa (Figure 1). We observed that the number of genes was differentially expressed in the eggs exhibiting increased efficiency for gynogenesis. Genes whose expression was correlated with the improved survival of gynogenetic progenies might be considered as candidate genes for gynogenesis in rainbow trout.

## 2. Materials and Methods

Ethical approval: this study was carried out in strict accordance with the recommendations in the Polish Act of 21 January 2005 of Animal Experiments (Dz. U. of. 2005 No 33, item 289). The protocol was approved by the Local Ethical Committee for Experiments on Animals of the University of Gdansk, Poland (no. 28/2015).

### 2.1. Gamete Collection

Rainbow trout utilized as gamete donors originated from the spring spawning broodstock raised in the Department of Salmonid Research, Inland Fisheries Institute in Olsztyn, Rutki, Poland. The number of dams and sires used in the research was determined to meet the requirements for the statistical analysis and it was based on the previous studies concerning the chromosomes set manipulations in fish [12,13,15]. Semen was taken from four males, pooled and kept in the plastic box. The motility of the spermatozoa was analyzed under microscope (Nikon Eclipse E 2000). Sperm was stored in the fridge until use. Four randomly selected females (F1–4) were stripped and the eggs were collected into separate plastic containers. After the quality examination, the eggs were kept at 10 °C until further manipulations. Fin clips from each gamete donors were stored in 96% ethanol for the molecular analysis. Approximately 240 eggs from each female were inseminated using untreated spermatozoa to establish control groups (Fc1–4) for the gynogenetic variants of the experiment.

### 2.2. Induction of Gynogenesis and Production of Doubled Haploids (DHs)

UV radiation was used to inactivate nuclear DNA in the rainbow trout spermatozoa. The portion of 0.375 mL of semen was diluted in 15 mL (40×) of seminal plasma. Then, 15.375 mL of diluted milt was exposed to the UV radiation for 11 min. During irradiation, the diluted milt was placed on the magnetic stirrer 20 cm under the UV-C lamp (30 W). After UV treatment, 1 µL sample of the milt mixed with 49 µL of sperm-activating medium (SAM) (154 mM NaCl and 1 mM Ca^2+^, buffered to pH 9.0 with 20 mM Tris + 30 mM glycine) [16] was checked under the microscope (Nikon Eclipse E 2000) to confirm the motility of irradiated spermatozoa.

For gynogenetic activation, approximately 980 eggs from each female were inseminated (activated) separately using irradiated sperm. About 3 min after activation, the eggs were washed with the hatchery water. To double haploid sets of maternal chromosome and to produce four stocks of gynogenetic doubled haploids (Fg1–4), gynogenetically activated eggs were exhibited to high hydrostatic pressure (HHP) shock (9500 PSI) 350 min after insemination. HHP shock lasted 3 min and was performed using TRC-APV electric/hydraulic apparatus (TRC Hydraulic Inc. Dieppe, Canada). Eggs were incubated in three replicates (Fg1–4) and in two replicates (Fc1–4) at 10 °C under routine conditions in the Department of Salmonid Research, Rutki.

### 2.3. Survival Rates

Survival rates of the gynogenetic doubled haploids and control fish were calculated 27 days post-fertilization (dpf) (eyed stage) and 57 dpf (swim-up stage). Differences in survivability were compared using Kruskal–Wallis test and estimated as significant when *p* < 0.05. All calculations were performed using Statistica software version 12 (StatSoft). All values in the text were expressed as average percentages (%) ± standard deviations (S.D.).

### 2.4. Molecular Analysis

Randomly selected larvae (n = 7–15) from each experimental group were sampled for the molecular analyses. Genomic DNA was extracted from the fins of parental and larvae tissues after hatching by using DNeasy Blood & Tissue Kit (Qiagen, Hilden, Germany) according to the manufacturer’s protocol. Nine polymorphic microsatellite markers including: OMM: 1025, 1053, 1075, 1084, 1101, 1107, 1127, 1131, 1238 [17,18] were used to genotype the gamete donors and gynogenetic progenies. Reaction mixtures were prepared as previously described [19].

The PCR reaction was accomplished using a Thermal Cycler SimpliAMP (Applied Biosystems, Foster City, CA, USA) according to the conditions of the PCR amplifications published in articles [18,19,20]. PCR products were electrophoresed in 1.5% agarose gel (Sigma-Aldrich, St. Louis, MO, USA) stained with ethidium bromide (0.05 mg/mL) and captured using a UV transilluminator (Vilber Laurmat ECX-20.M, Eberhardzell, Germany).

### 2.5. Transcriptome Analysis

In the present research, we used the transcriptome data (normalized reads counts) already used to analyze the changes in the transcriptome of non-activated eggs, irradiated and fertilized eggs (androgenesis) and the eggs activated with UV-irradiated spermatozoa (gynogenesis) [20]. Though, in that already published report, we did not consider the inter-clutch transcriptome differences which could largely contribute to the eggs’ survival following gynogenesis [20].

The methods of RNA extraction, purification, library construction and the whole transcriptome sequencing of rainbow trout eggs have been already described [20]. Shortly; four randomly chosen eggs from each batch were preserved in RNA-later solution incubated overnight in a refrigerator (4 °C) and finally stored at −80 °C. The eggs were thawed on ice and manually homogenized in TRIzol Reagent (Thermo Fisher Scientific Waltham, MA, USA) to isolate the RNA. Before the library construction, the total RNA was additionally purified using Agencourt RNAClean XP beads (Beckman Coulter, Brea, CA, USA). A total amount of 800 ng purified RNA was used as input for TruSeq RNA Sample Prep v2 kit (Illumina). Standard library construction steps were followed by a qualitative (Agilent TapeStation 2200, Agilent, Stockport, UK) and quantitative (Qubit, Thermo Fisher Scientific, Waltham, MA, USA) evaluation. Quality controlled and normalized libraries were eventually sequenced in a single 50 bp run (1 × 50 bp) on the HiScanSQ system using TruSeq SBSv3 Sequencing kit (Illumina, San Diego, CA, USA) to provide approximately 25 million reads per sample. Provided demultiplexed raw reads were controlled for quality using FastQC software and filtered using Flexbar software [21]. The high quality reads were mapped against the newest available and supplemented rainbow trout transcriptome (encompassing 44990 transcripts; downloaded from http://www.animalgenome.org/repository/pub/MTSU2014.1218/ [22] using Bowtie software [23]. The mapped reads were analyzed using eXpress (http://bio.math.berkeley.edu/eXpress/overview.html; [24]). The obtained estimated rounded effective read counts for the separate transcripts and samples were counted using DESeq2 [25]. Principal components analysis (PCA) as a singular value decomposition with imputation and heatmap generation along with hierarchical clustering based on Euclidean distance were performed using ClustVis software [26].

Transcriptome data from the control and gynogenetic eggs were studied in a way that allowed us to correlate genes’ expression with gynogenesis efficiency measured by the average eggs survival rates. We calculated the correlation coefficients between the level of expression of each analyzed transcript detected by RNA-Seq approach and the survival rate of eggs obtained from the separate females. Correlation coefficients were calculated for the expression of genes from the non-activated eggs (control groups) as well as in eggs activated by the UV-irradiated spermatozoa (gynogenetic groups). To identify the genes that were associated with the efficiency of gynogenesis, the transcripts expressed with a base mean > 1 and the sum of correlation coefficients absolute values > 1.99 in eggs before and after activation with UV-treated sperm were selected. The most strongly correlating transcripts were annotated and functionally analyzed using Blast2GO software considering gene ontology (GO) terms.

## 3. Results

### 3.1. Survival Rates and Genetic Analysis Examination

The survival rates of rainbow trout from control groups (Fc1–Fc4) varied between 91.9 ± 0.72% and 96.4 ± 1.46% at the eyed stage. At the swim-up stage, survivability decreased slightly and varied between 85.6 ± 6.23 and 96.4 ± 1.46%. The survivability of the DHs in groups Fg1, Fg3, Fg4 at the eyed stage equaled 3.5 ± 0.99%, 3.1 ± 0.76% and 6.2 ± 3.70%, respectively. At the swim-up stage, the survival rates decreased to 0.8 ± 0.90%, 1.0 ± 0.34% and 5.0 ± 1.96%, respectively. The gynogenetic DHs from Fg2 group exhibited an extremely high survival rate at the eyed stage (57.2 ± 2.95%). At the swim-up stage, the survival rate in this variant was lower but still very high (41.5 ± 2.10%) (*p* < 0.05) (Table 1).

All the studied egg donors were heterozygous in each studied loci, whereas all the gynogenetic progenies were homozygous (Appendix A).

### 3.2. Detection of Genes Associated with Efficiency of Gynogenesis 

To detect the genes potentially associated with the efficiency of gynogenesis, we correlated the expression of individual transcripts in individual females with survival rates observed for gynogenetically activated eggs. Applied genes filtering criteria allowed to detect 63 transcripts that belonged to 46 different genes that altered the expression (mostly up-regulation) strongly correlated with survival in both, control as well in gynogenetically activated eggs (Appendix A). The applied hierarchical clustering of the transcripts’ expression allowed for the clear separation of both, control and gynogenetic groups. In both the control and gynogenetic group, the sample from one female (namely no 2.) showed a strongly divergent expression pattern (Figure 2).

Gynogenetic embryos developing in the eggs from female no. 2 were shown to have a significantly higher survival rate than the embryos developing in the eggs from the remaining females, which suggested that the altered transcripts which were found in these eggs can contribute to the observed differences in the efficiency of gynogenesis. The functional analysis of the these transcripts showed their engagement, meanly in the biological processes associated with: developmental processes (e.g., anatomic structure development, genes like: homeobox protein Nkx-2.2; TBC1 domain family member 24-like; tyrosine 3-monooxygenase), differentiation (e.g.,: contactin-4-like isoform X1; transcription factor SOX-11-like; tyrosine 3-monooxygenase) and cellular metabolic processes (like e.g.,: elongator complex protein 5, forkhead box protein N4-like, telomerase reverse transcriptase, forkhead box protein O4). The genes were also connected with cellular components such as cell membrane (carnitine O-palmitoyltransferase 1) and nucleus and were primarily associated with molecular functions such as: binding (especially DNA binding) and catalytic activity (Appendix A).

## 4. Discussion

The application of the completely homozygous gynogenetic doubled haploids (DH) enables studies on the phenotypic consequences of the recessive alleles and improves the de novo assembly of the genomes sequenced using next-generation sequencing approach. Gynogenetic DHs are used in the selective breeding programs and allowed for the generation of fish clonal lines that have been established in several aquaculture and model fish species [16]. Unfortunately, the potential application of DHs in developmental research and aquaculture is limited by a rather low survival rate of such specimens that rarely exceeds 15% at the swim-up stage [8]. The expression of lethal alleles, the side effects of the egg treatment after activation (physical shock) and low egg quality are listed as factors that are responsible for the high mortality of the homozygous gynogenotes [8].

In the present research, we observed inter-clutch differences in the survival of gynogenetic DHs. The gynogenetic offspring of three females show similar survivability that did not exceed 5% at the swim-up stage. However, the survival of gynogenetic larvae that hatched from the eggs originating from the fourth female (F_2_) was almost 10 times higher (Table 1). Taking into account the fact that the genome of gynogenetic specimens was made exclusively of maternal chromosomes, inter-clutch differences in the survival of gynogenetic offspring may result from the inter-individual differences. Differences in the survival of gynogenetic individuals developing in eggs from different females may be related to the inter-individual variation in the egg quality and the genetic differences between the egg donors. Female F_2_ might have had less recessive lethal alleles than three other egg donors, which would result in the lower mortality of its DH progenies. The eggs of different origin showing different quality may have also altered transcriptome profiles [3]. When assessing the influence of HHP on the transcriptome of the rainbow trout eggs, it was noticed that the differences in the maternal gene expression were greater between the eggs from different clutches than between the HHP-treated and the untreated eggs which originated from an individual female [12]. It was presumed that the inter-individual differences between rainbow trout females from the same broodstock had a larger influence on the egg transcriptome and the developmental ability than the HHP treatment of the eggs. In zebrafish, the differences in the expression of maternal genes in eggs from different clutches reflected differences between mothers [11].

In fish, RNA-seq analysis has been used to study inter and intra-individual transcriptomic differences and to discover the candidate genes for diverse research topics, including different reproductive modes such as natural gynogenesis [27] and induced parthenogenesis [28], among others. Here, the preliminary results of comparative transcriptome analysis evidenced inter-clutch differences and the eggs from one of the females (F_2_) exhibited a different expression patter of 46 genes when compared to the eggs from three other females (Figure 1, Appendix A). Although, only a limited number of egg donors was studied and no quantitative reverse transcription PCR (RT-qPCR) was performed, these genes may be considered as potential candidate genes whose expression in eggs affect their competence for gynogenetic development. Our previous study showed that the obtained RNA-Seq results were sufficiently accurate to evaluate global changes in the gene expression, as well as the expression level of individual genes and their isoforms as they were positively validated by RT-qPCR [21]. That is why no confirmatory validation was applied in this study.

Eggs with the highest survival of gynogenetic embryos showed an up-regulation of genes that are associated with processes influencing egg quality and development capacity including cell survival, migration and differentiation (tyrosine-protein kinase receptor TYRO3-like gene), triglyceride metabolism (carnitine O-palmitoyltransferase 1), gene silencing and early embryogenic development (protein argonaute-3 gene, leucine-rich repeat-containing protein 3-like), 5S RNA binding (ribosome biogenesis regulatory protein homolog), transcriptional factor involved in the embryonic neurogenesis and tissue modelling during development (transcription factor SOX-11-like gene), encoding enzyme that participates in polyunsaturated fatty biosynthesis (3-oxoacyl-acyl-carrier-protein reductase gene) as well as senescence and aging (telomerase reverse transcriptase, *TERT* gene).

The TERT gene seems to be particularly interesting in the context of fertility, egg quality and embryonic development, including parthenogenesis. TERT is a protein catalytic subunit of telomerase, an enzyme that maintains the length of the telomeric DNA. Contrary to mammals where TERT expression and telomerase activity are restricted to germ line cells, stem cells and tumor cells, fish including rainbow trout show a high activity of telomerase in somatic tissues regardless of the age of the studied specimens [29]. In *Xiphophorus maculatus* and *X. helleri*, the expression of TERT in ovary and testis was significantly higher than in other organs [30]. Moreover, in killifish (*Fundulus heteroclitus*) the TERT expression was confirmed in eggs [31]. Male TERT^-/-^ zebrafish mutants exhibit a significantly decreased production of mature sperm while TERT^-/-^ females become infertile with age [32]. Progenies of TERT^-/-^ parents die early during embryogenesis. Dramatic reduction in fertility is observed in TERT-deficient killifish males. In turn, TERT mutant females have atrophied ovaries, premature defects in the germline and lay fewer eggs than the females with TERT expression [33]. In some mammalian species, telomerase is required for oocyte development and the decline in TERT expression is paralleled with a decline in oocyte quality [34]. In mice, the significant decreasing of TERT abundance in oocytes is observed during reproductive and postovulatory aging [35]. Moreover, telomerase activity has been confirmed to be important during parthenogenesis [34].

## 5. Conclusions

Rainbow trout eggs with the highest survival of gynogenetic specimens also exhibited an increased expression of 46 genes. These genes are associated with cell survival, migration and differentiation, triglyceride metabolism, the biosynthesis of polyunsaturated fat, early embryogenic development, 5S RNA binding, embryonic neurogenesis and tissue modelling during development as well as senescence and aging, among others. Identified genes may be considered as potential candidate genes whose expression is correlated with the efficiency of gynogenesis in rainbow trout. Special attention has been dedicated to the TERT gene that maintains the length of the telomeric DNA and its high expression exhibited in the rainbow trout eggs with high developmental capacity for gynogenesis suggests that telomerase may also be involved in the processes related to egg quality, early ontogeny and gynogenetic development in fish.

## Figures and Tables

**Figure 1 animals-10-01326-f001:**
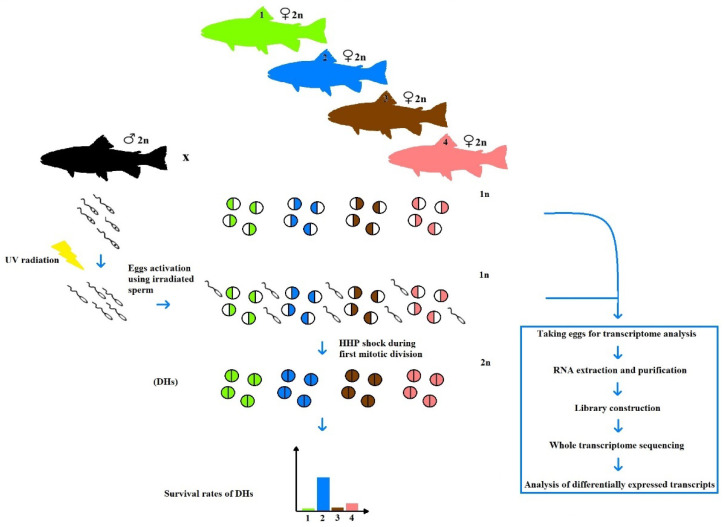
Scheme of the experimental design to produce gynogenetic doubled haploids (DHs) in rainbow trout and to perform the transcriptome analysis of eggs that show different ability for gynogenetic development. UV-irradiated spermatozoa were used to activate eggs and high hydrostatic pressure (HHP) shock was applied to duplicate maternal chromosomes.

**Figure 2 animals-10-01326-f002:**
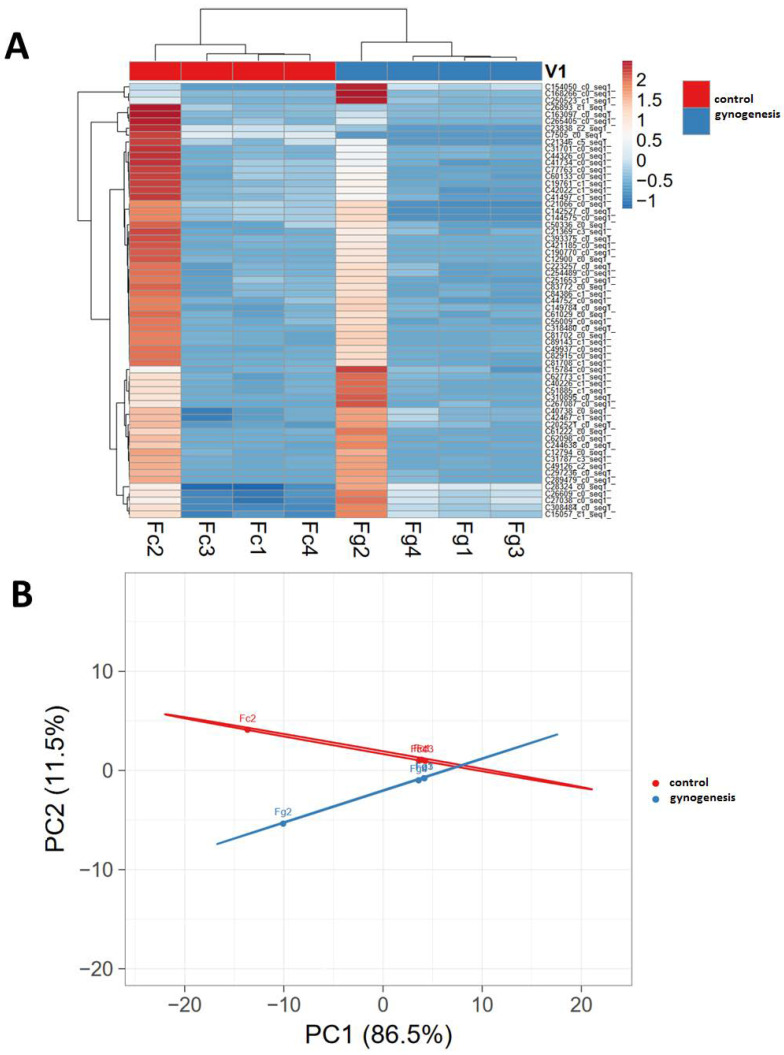
Hierarchical clustering (**A**) and principal components analysis (**B**) of the expression of the transcripts that were the most strongly correlated with the efficiency of gynogenesis. Fc1–Fc4–transcripts from the non-activated eggs (control) originated from F1–F4 females, respectively, and Fg1–Fg4–transcripts from gynogenetically activated eggs (gynogenesis) originated from the same females.

**Table 1 animals-10-01326-t001:** Survival rates (averages ± standard deviations (SD)) of the rainbow trout from the control (C) and the diploid gynogenetic doubled haploid (DHs) groups.

Variants of Experiment	Eyed Stage (%)	Swim-Up Stage (%)
Control (C)		
Fc1	95.0 ± 4.70	88.7 ± 7.63
Fc2	94.5 ± 0.67	85.6 ± 6.23
Fc3	91.9 ± 0.72	89.8 ± 0.42
Fc4	96.4 ± 1.46	96.4 ± 1.46
Gynogenesis (DHs)		
Fg1	3.5 ± 0.99	0.8 ± 0.90
Fg2	57.2 ± 2.95	41.5 ± 2.10
Fg3	3.1 ± 0.76	1.0 ± 0.34
Fg4	6.2 ± 3.70	5.0 ± 1.96

## Data Availability

Data available on request due to privacy/ethical restrictions.

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
