# Peer review of "Preliminary Identification of Candidate Genes Related to Survival of Gynogenetic Rainbow Trout (Oncorhynchus mykiss) Based on Comparative Transcriptome Analysis"

_animals, 2020, doi:10.3390/ani10081326_

Round 1

Reviewer 1 Report

This study done by Ocalewicz et al use comparative transcriptomic approach on testing the hypothesis that the high variation of success rate for gynogenesis in rainbow trout is correlated to their egg quality. To test this hypothesis, they performed RNAseq and gene mapping to perform gene expressional profiling. Eventualy, they narrow down to less than 50 genes that might be associated with the high egg quality variation. Basically, I perfsonally think this research is important for aquaculture fields and also provide some answers to the question why gynogenesis sucess rate gets so high variation at molecular level. Therefore, I favor this paper be published in Animals, but still need to do some major revision to clarify some unclear issues.

Major issues:

  1. They did not perform RT-PCR validation experiemnt. Therefore, it is difficult to validate the accuracy of marker gene up- and down-regulated. The author has to consider seriously how to validate their finding by doing RT-PCR, or have to discuss the potential weakness of their study due to lack of RT-PCR validation
  2. The high sucess rate for gynogenesis from female 2 is interesting. efore, it is difficult to validate the accuracy of marker gene up- and down-regulated. The author has to consider seriously how to validate their finding by doing RT-PCR, or have to discuss the potential weakness of their study due to lack of RT-PCR validation
  3. The high sucess rate for gynogenesis from female 2 is interesting. I  am wondering could it possible to provide other parameters to evaluate the egg quality in addition to NGS data ? For example, to check the mitochondrial content or lipid/nutrition content inside this batch eggs. 
  4. The author can also do comparative study by comparing the NGS or microaary data from other fish species (like zebrafish), to see whether any candidate marker genes showing similar pattern with rainbow trout. Or the author can try to do some discussion in order to strngth their poit-of-view.
  5. The figure legend is too simple. The author need to do more writing to provide more details. For ecample, what is the ma=eaning for label 0_2, 0_1, ... and so on. 
  6. The authors have considered to provide a cartoon picture as Figure 1 to illustarte their entire exprerimental design and workflow to generate gynogenesis rainbow trout and the following RNAseq experiments. This picture can provide more informative and attracktive information to the audiences.
  7. The most important marker they proposed is tert gene. The author have to validate the expression of this gene to provide more solid conclusion.   
  8. The author have to provide the data matrix in excel format to perform PCA and clustering as supplymentary data. This information is important to let the audience to do double validation jobs.                  

Author Response

Answers for the reviewers’ comments and suggestions

Reviewer 1

Konrad Ocalewicz (KO): Thank you for the comments and suggestions on the submitted manuscript. Below are our answers . The entire text has been improved and the changes in the text are easy to find as track changes function was On.

This study done by Ocalewicz et al use comparative transcriptomic approach on testing the hypothesis that the high variation of success rate for gynogenesis in rainbow trout is correlated to their egg quality. To test this hypothesis, they performed RNAseq and gene mapping to perform gene expressional profiling. Eventualy, they narrow down to less than 50 genes that might be associated with the high egg quality variation. Basically, I perfsonally think this research is important for aquaculture fields and also provide some answers to the question why gynogenesis sucess rate gets so high variation at molecular level. Therefore, I favor this paper be published in Animals, but still need to do some major revision to clarify some unclear issues.

Major issues:

  1. They did not perform RT-PCR validation experiemnt. Therefore, it is difficult to validate the accuracy of marker gene up- and down-regulated. The author has to consider seriously how to validate their finding by doing RT-PCR, or have to discuss the potential weakness of their study due to lack of RT-PCR validation

KO: In our study, all steps of mRNA and miRNA sequencing were added to the strict control. The input RNA quality was high and produced libraries gave satisfying QC metrics. Also other parameters, like reads quality, mapping efficiency etc. were of good quality, so we are convinced that our results are accurate as possible. Several previous studies have shown that RNA-Seq results are sufficiently accurate to evaluate global changes in genes expression as well as expression level of individual genes and their isoforms. In case of working with draft genome sequences and computationally reconstructed transcriptomes, qPCR validation might be tricky and may not give correct answers. Transcriptome sequence required for primers design may not include all transcript isoforms and thus gene expression analysis will be biased by the lack of quantification of some splice variants with qPCR which were counted using RNA-Seq. So, we anticipate that both RNA-Seq and qPCR results might be biased due to the still poor annotation of Rainbow Trout genome. Another, the most limiting factor is the fact that fund for this project are already utilized, thus not allowing us to perform any additional molecular analysis.

  1. The high sucess rate for gynogenesis from female 2 is interesting. efore, it is difficult to validate the accuracy of marker gene up- and down-regulated. The author has to consider seriously how to validate their finding by doing RT-PCR, or have to discuss the potential weakness of their study due to lack of RT-PCR validation

KO: this has been discussed -  ll. 264-274, ll. 234-237, ll. 277-279

  1. The high sucess rate for gynogenesis from female 2 is interesting. I  am wondering could it possible to provide other parameters to evaluate the egg quality in addition to NGS data ? For example, to check the mitochondrial content or lipid/nutrition content inside this batch eggs. 

KO: Actually, for the trout eggs there are no solid markers that enable evaluation of their quality for gynogenesis. The best is to analyze survival of eggs. Here, survival of control fish here was very high what confirmed used eggs were of the highest quality but in gynogenetic variant of the experiment it was observed that good quality eggs are not always good enough for gynogenesis.

  1. The author can also do comparative study by comparing the NGS or microaary data from other fish species (like zebrafish), to see whether any candidate marker genes showing similar pattern with rainbow trout. Or the author can try to do some discussion in order to strngth their poit-of-view.

KO: To our knowledge, this is the first report concerning identification of differentially expressed genes in gynogenetically activayed fish eggs. But we found two papers that used NGS to study gene transcripts in organisms that reporduce using natural gynogenesis and parthenogensis (ll. 264-267)

  1. The figure legend is too simple. The author need to do more writing to provide more details. For ecample, what is the ma=eaning for label 0_2, 0_1, ... and so on. 

KO: The figure and the legend were updated and corrected. Now it is Figure 2 as we have provided Figure 1 with the experimental design.

  1. The authors have considered to provide a cartoon picture as Figure 1 to illustarte their entire exprerimental design and workflow to generate gynogenesis rainbow trout and the following RNAseq experiments. This picture can provide more informative and attracktive information to the audiences.

KO: We have provided such figure,. It is Figure 1.

  1. The most important marker they proposed is tert gene. The author have to validate the expression of this gene to provide more solid conclusion. 

KO: We will do this when we receive funds for another project that will be focused on teh selected genes provided by NGS approach described here.  However, it has been also discussed in the ms: 234-237, ll. 277-279

  1. The author have to provide the data matrix in excel format to perform PCA and clustering as supplymentary data. This information is important to let the audience to do double validation jobs. 

KO: In the supplementary materials these data are in the Excel file.              

Reviewer 2 Report

The eggs from different fish individuals used for gynogenesis showed vast difference in survival. I also performed gynogenesis in fish and the problem confused me as well. The present study analyzed the transcriptome of eggs showing vast differences in survivals of gynogenesis. My concern is whether transcriptome data was from the eggs used in control and production of doubled haploids. On line 150-154, the description was very confused and please clarify it. If not, I am afraid the present data could not support the conclusion. Some errors were listed as following:

  1. Change the title to “Comparative transcriptome analysis reveals potential candidate genes related to survival of gynogenesis in rainbow trout (Oncorhynchus mykiss)”.
  2. Line 19, “differences” or “difference”?
  3. Line 151-154, As mentioned above, please clarify the samples for RNA-Seq analysis.
  4. Line 190, The “p” should be “italic”.
  5. Table 1, Change “91,6”to “91.6”.
  6. Fig.1 Give a detailed figure caption.
  7. In the transcriptome data, I wonder the gene expression profiling in the eggs showing low survival of gynogenesis. I suggest the authors further analyzed the transcriptome data to address the issues.

Author Response

Reviewer 2

Konrad Ocalewicz (KO): Thank you for the comments and suggestions on the submitted manuscript. Please, find below our answers . The entire text has been improved and the changes in the text are easy to find as track changes function was On.

The eggs from different fish individuals used for gynogenesis showed vast difference in survival. I also performed gynogenesis in fish and the problem confused me as well. The present study analyzed the transcriptome of eggs showing vast differences in survivals of gynogenesis. My concern is whether transcriptome data was from the eggs used in control and production of doubled haploids. On line 150-154, the description was very confused and please clarify it. If not, I am afraid the present data could not support the conclusion. Some errors were listed as following:

  1. Change the title to “Comparative transcriptome analysis reveals potential candidate genes related to survival of gynogenesis in rainbow trout (Oncorhynchus mykiss)”.

KO: The title has been changed

  1. Line 19, “differences” or “difference”?

KO: This has been corrected.

  1. Line 151-154, As mentioned above, please clarify the samples for RNA-Seq analysis.

KO: In the revised version of ms this part has been re-worded to clarify RNA seq part. Ll. 152-158.

  1. Line 190, The “p” should be “italic”.

KO: It has been changed.

  1. Table 1, Change “91,6”to “91.6”.

KO: this one has been also corrected.

  1. Fig.1 Give a detailed figure caption.

KO: In the revised version of ms, this is Figure 2 that caption is improved.

  1. In the transcriptome data, I wonder the gene expression profiling in the eggs showing low survival of gynogenesis. I suggest the authors further analyzed the transcriptome data to address the issues.

KO: The further analysis of particular genes will be possible when we receive new funds as project that enabled us to provide results described in this paper is finished. ll. 264-274, ll. 234-237, ll. 277-279

Reviewer 3 Report

The study includes insuffifient number of replicates - only single fish is used as a representat of high survival following gynogenesis. The MS and overall concept is scientifically sound, though the chance of describing biological artifact is pretty high what disqualifies ms from publication, in my oppinion. I understand the value of finding such a fish, but the data based on single fish is just not enough, to me. 

I suggest the Authors to try to generate more data, such as real-time qPCR validation of candidate genes on higher number of eggs with various quality following gynogenesis and try to make correlation or statistically sound analysis. The best would be - though - validate candidate genes on low- and high-survival eggs following gyogenesis, but I assume that this would be simply very difficult as this single fish (found in this study) is actually an artifact. 

Author Response

Reviewer 3

Konrad Ocalewicz (KO): Thank you for the comments and suggestions on the submitted manuscript. Please, find attached our answers . The entire text has been improved and the changes in the text are easy to find as track changes function was On.

The study includes insuffifient number of replicates - only single fish is used as a representat of high survival following gynogenesis. The MS and overall concept is scientifically sound, though the chance of describing biological artifact is pretty high what disqualifies ms from publication, in my oppinion. I understand the value of finding such a fish, but the data based on single fish is just not enough, to me. 

KO: from our experience, such a high survival of gynogenetic trout has been observed in the studied stock earlier as we have performer gynogenesis for several years to produce all-female stocks and fish for hormonal masculinization. It is rare that true but I would not describe it as a biological artifact. In the future research we would like to perform gynogenesis using much more gamete donors to screen the stock looking for females that eggs show increased capacity for gynogenesis as described in this manuscript. The qPCR will be used to examine each of the 46 genes described here. But this needs a new project/funds for at least  two years as rainbow trout stock is a spring spawning one.

I suggest the Authors to try to generate more data, such as real-time qPCR validation of candidate genes on higher number of eggs with various quality following gynogenesis and try to make correlation or statistically sound analysis. The best would be - though - validate candidate genes on low- and high-survival eggs following gyogenesis, but I assume that this would be simply very difficult as this single fish (found in this study) is actually an artifact. 

KO: We totally agree that it would nice to do a qPCR but here we do not have any funds to perform RT-PCR, secondly we do not have enough time to design actually a new experiment (only 10 days for the revision of the paper). So, the further analysis of particular genes will be possible when we receive new funds as project that enabled us to provide results described in this paper is finished. Some parts of the revised ms is dedictaed to this issue;   ll. 264-274, ll. 234-237, ll. 277-279

Round 2

Reviewer 1 Report

In this revised version, the author already did adequte response for my suggestion. Now this paper is suitable been published in Animals. The detail information of tool and method use to generate heat map and PCA should be updated in the materials and methods.  

Author Response

Comments and Suggestions for Authors

In this revised version, the author already did adequte response for my suggestion. Now this paper is suitable been published in Animals. The detail information of tool and method use to generate heat map and PCA should be updated in the materials and methods.  

Konrad Ocalewicz: Principal components analysis (PCA) as a singular value decomposition with imputation and heatmap generation along with hierarchical clustering based on Euclidean distance were performed using ClustVis software. ll 182-184, reference no 27. 

Thank you for your comments and suggestions that were very helpful.

Reviewer 2 Report

I am satisfied with the revision by the authors. The manuscript could be accepted at this status. 

Author Response

Comments and Suggestions for Authors

I am satisfied with the revision by the authors. The manuscript could be accepted at this status. 

Konrad Ocalewicz: Thank you for your help to improve this paper.

Reviewer 3 Report

The MS is still suffering of methodical flaw - usage of single individual for transcriptomic analysis without further validation. To me this is unacceptable for scientific publication. 

The authors claimed that they are going to perform further studies using qPCR for validation of the gene set identified on eggs coming from more females. If yes - that is fantastic. Then, they will be able to publish the results presented in this MS along with the new data. By now, I have no other option but to sustain my oppinion. 

Author Response

Dear Reviewer,

Please find below an answer for your comment on the revised paper.

The MS is still suffering of methodical flaw - usage of single individual for transcriptomic analysis without further validation. To me this is unacceptable for scientific publication. 

The authors claimed that they are going to perform further studies using qPCR for validation of the gene set identified on eggs coming from more females. If yes - that is fantastic. Then, they will be able to publish the results presented in this MS along with the new data. By now, I have no other option but to sustain my oppinion. 

Konrad Ocalewicz: in this paper we have proposed a limited number of candidate genes that expression may be related to efficiency of gynogenesis in rainbow trout what in my opinion and opinions of two reviews who evaluated this paper has itself some scientific value. The title and the content of the paper point that the provided results are rather first step to find genes related to gynogenesis in fish rather than a finished work or the final composition. The second step requires another gynogenetic experiment , screening of eggs showing varied ability for gynogenesis, and qPCR verification of chosen candidate genes. As examined rainbow trout represent Spring spawning stock all together will take more than one additional year work and investment of substantial funds. Thus, I believe this paper – a first step in the research concerning molecular aspects of gynogenesis in fish is worth publishing  to give an introduction for the second step.

Beste wishes,